# Uptake-Dependent and -Independent Effects of Fibroblasts-Derived Extracellular Vesicles on Bone Marrow Endothelial Cells from Patients with Multiple Myeloma: Therapeutic and Clinical Implications

**DOI:** 10.3390/biomedicines11051400

**Published:** 2023-05-08

**Authors:** Aurelia Lamanuzzi, Ilaria Saltarella, Antonia Reale, Assunta Melaccio, Antonio Giovanni Solimando, Concetta Altamura, Grazia Tamma, Clelia Tiziana Storlazzi, Doron Tolomeo, Vanessa Desantis, Maria Addolorata Mariggiò, Jean-François Desaphy, Andrew Spencer, Angelo Vacca, Benedetta Apollonio, Maria Antonia Frassanito

**Affiliations:** 1Unit of Internal Medicine and Clinical Oncology, Department of Precision and Regenerative Medicine and Ionian Area, University of Bari Aldo Moro, 70124 Bari, Italy; 2Unit of Pharmacology, Department of Precision and Regenerative Medicine and Ionian Area, University of Bari Aldo Moro, 70121 Bari, Italy; 3Myeloma Research Group, Australian Centre for Blood Diseases, Central Clinical School, Monash University-Alfred Health, Melbourne, VIC 3004, Australia; 4Department of Biosciences, Biotechnologies and Environment, University of Bari Aldo Moro, 70121 Bari, Italy; grazia.tamma@uniba.it (G.T.);; 5Unit of Clinical Pathology, Department of Precision and Regenerative Medicine and Ionian Area, University of Bari Aldo Moro, 70121 Bari, Italy; 6Malignant Haematology and Stem Cell Transplantation, Department of Haematology, Alfred Hospital, Melbourne, VIC 3004, Australia; 7Department of Clinical Hematology, Monash University, Melbourne, VIC 3004, Australia

**Keywords:** angiogenesis, angiogenic cytokines, extracellular vesicles, multiple myeloma

## Abstract

Extracellular vesicles (EVs) have emerged as important players in cell-to-cell communication within the bone marrow (BM) of multiple myeloma (MM) patients, where they mediate several tumor-associated processes. Here, we investigate the contribution of fibroblasts-derived EVs (FBEVs) in supporting BM angiogenesis. We demonstrate that FBEVs’ cargo contains several angiogenic cytokines (i.e., VEGF, HGF, and ANG-1) that promote an early over-angiogenic effect independent from EVs uptake. Interestingly, co-culture of endothelial cells from MM patients (MMECs) with FBEVs for 1 or 6 h activates the VEGF/VEGFR2, HGF/HGFR, and ANG-1/Tie2 axis, as well as the mTORC2 and Wnt/β-catenin pathways, suggesting that the early over-angiogenic effect is a cytokine-mediated process. FBEVs internalization occurs after longer exposure of MMECs to FBEVs (24 h) and induces a late over-angiogenic effect by increasing MMECs migration, chemotaxis, metalloproteases release, and capillarogenesis. FBEVs uptake activates mTORC1, MAPK, SRC, and STAT pathways that promote the release of pro-angiogenic cytokines, further supporting the pro-angiogenic milieu. Overall, our results demonstrate that FBEVs foster MM angiogenesis through dual time-related uptake-independent and uptake-dependent mechanisms that activate different intracellular pathways and transcriptional programs, providing the rationale for designing novel anti-angiogenic strategies.

## 1. Introduction

Multiple myeloma (MM) is a plasma cell malignancy originating in the bone marrow (BM), typically preceded by asymptomatic premalignant conditions called monoclonal gammopathy of undetermined significance (MGUS) and smoldering myeloma (SMM) [1].

The crosstalk between tumor plasma cells (MM cells) and the BM microenvironment is essential for transitioning from the premalignant phases to full-blown MM. MM cells hijack immune and non-immune components of the BM to sustain tumor cell survival, proliferation, invasion, and immune escape [2]. Among the numerous defective features of the BM milieu, aberrant angiogenesis is a prominent hallmark of MM pathogenesis and has been correlated to progression and resistance to therapy [3,4].

Many standard anti-myeloma drugs currently used in clinical practice (i.e., bisphosphonates, proteasome inhibitors, and immunomodulatory drugs (IMiDs)) have overlapping anti-angiogenic effects in vitro [5], consolidating the promising potential of angiogenesis-targeting approaches in combined regimens. So far, different therapies targeting angiogenesis have been developed and applied in MM settings. However, they have shown only limited therapeutic efficacy [5]. Therefore, a deeper understanding of the biological processes fostering aberrant BM angiogenesis is needed to improve the efficacy of anti-angiogenic drugs in MM [6].

Endothelial cells (ECs) isolated from MM patients (MMECs) display an activated phenotype with an increased pro-angiogenic capability both in vitro and in vivo [3]. Several mechanisms confer an over-angiogenic phenotype to MMECs, including the deregulated secretion of cytokines and growth factors, altered pathways activation, imbalanced non-coding RNAs production, and extracellular vesicles (EVs) release [4,7,8,9,10,11].

Aberrant angiogenesis is supported not only by MM cells but also by BM stromal cells (BMSCs). Among them, MM fibroblasts (MM FBs) secrete soluble factors (SDF1α, VEGF, and FGF2) with pro-angiogenic potential both in vitro and in vivo [12]. Like other cells, FBs communicate with the surrounding environment through EV secretion [13]. However, the role of FB-derived EVs (FBEVs) in modulating BM angiogenesis of MM patients has not been studied yet.

Based on our previous study demonstrating the pro-angiogenic potential of MM FBs [12], we investigate the contribution of FBEVs in supporting BM angiogenesis. We demonstrate that FBEVs’ cargo contains several angiogenic cytokines that foster an early pro-angiogenic effect, independent from EVs uptake, via the activation of the VEGF/VEGFR2, HGF/HGFR, and ANG-1/Tie2 axis. On the other hand, FBEVs internalization, which occurs after a prolonged exposure, modulates additional angiogenic abilities of MMECs by activating different intracellular pathways.

Overall, our data indicate that FBEVs induce a dual pro-angiogenic response: an early uptake-independent angiogenic effect and a late uptake-dependent response. Our results provide additional information on the EVs-mediated mechanisms leading to enhanced angiogenesis in the MM BM microenvironment.

## 2. Materials and Methods

### 2.1. Patients

Twenty-one patients fulfilling the International Myeloma Working Group (IMWG2014 diagnostic criteria for symptomatic MM [14]) were studied. The study was approved by the Ethics Committee of the University of Bari Medical School (ID No. 1335/2015), and all patients provided their informed consent in accordance with the Declaration of Helsinki. 

### 2.2. Cell Cultures

Bone marrow-derived primary ECs and FBs from MM patients were isolated from fresh BM aspirates, as previously described [10,12]. Briefly, BM mononuclear cells were obtained by centrifugation on Ficoll–Hypaque gradient (Pharmacia Biotech, Uppsala, Sweden). Cells were then cultured in RPMI 1640 medium supplemented with 10% fetal bovine serum (FBS) and 1% penicillin/streptomycin (Euroclone, Milan, Italy) until confluence. Adherent BM mononuclear cells were detached, and MM FBs and MMECs were then purified by D7-FIB-conjugated and anti-CD31-conjugated microbeads (Miltenyi Biotech, Bergisch Gladbach, Germany). Cell purity was assessed by flow cytometry. MM FBs and MMECs were cultured in DMEM (Lonza, Basel, Switzerland) supplemented with 1% penicillin/streptomycin (Euroclone) and 20% FBS (Sigma-Aldrich, Saint Louis, MO, USA) and routinely tested for mycoplasma contamination (PlasmoTest^TM^; InvivoGen, San Diego, CA, USA).

### 2.3. Reagents

The inhibition of VEGFR2, HGFR and Tie2 receptors was performed by using neutralizing antibodies against VEGFR2 (100 ng/mL), HGFR (10 µg/mL) and Tie2 (10 µg/mL) all from Biotechne. MMECs were pre-incubated with blocking antibodies for 30 min at 37 °C and then co-cultured with FBEVs for the indicated time.

### 2.4. EVs purification, Characterization and Co-Culture

The MM FBEVs were purified from MM FBs cultured for 72 h in DMEM medium (Sigma-Aldrich) supplemented with 10% of exosome-depleted FBS (Euroclone) as described [15]. Briefly, cell culture media were sequentially centrifuged at increasing speeds to remove dead cells and debris, filtered with 0.22 μm pore filters (Merck Millipore, Darmstadt, Germany) to remove small debris and larger vesicles, and concentrated with a protein concentrator (Ultra-15 MWCO 10 kDa; Merck Millipore, Burlington, MA, USA). EVOs were precipitated using ExoQuick-TC exosome precipitation solution (SBI System Biosciences, Palo Alto, CA, USA) according to the manufacturer’s instructions. Purified FBEVs were characterized by flow cytometry using the Megamix-Plus SSC kit (BioCytex, Marseille, France) according to the manufacturer’s instructions. The protein concentration of each EVs pellet was quantified using the Bradford assay (Bio-Rad Laboratories, Hercules, CA, USA). Co-culture experiments were performed by adding 400 µg of FBEVs to 4 × 10^4^ MMECs.

### 2.5. EVs Uptake

FBEVs were labeled with the cell membrane tracker BODIPY TR™ ceramide, according to the manufacturer’s instructions (Thermo Fisher Scientific, Waltham, MA, USA), and co-cultured with MMECs for 3, 6, 12, and 24 h. The uptake of EVs was monitored using FACScanto II (BD Biosciences, Franklin Lakes, NJ, USA). Data analysis was performed with FlowJo v.10.

### 2.6. Immunofluorescence–Confocal Laser-Scanning Microscopy

MMECs were co-cultured with FBEVs labelled with the cell membrane tracker BODIPY TR ceramide (Thermo Fisher Scientific) for 3, 6, 12, and 24 h. Next, MMECs were fixed with 4% paraformaldehyde, permeabilized with Triton-X100 and then incubated with phalloidin-FITC (Sigma-Aldrich) for F-actin staining. Images were acquired with a confocal laser-scanning fluorescence microscope Leica TCS SP2 (Leica Microsystems, Heerbrugg, Switzerland) as previously described [16].

### 2.7. Capillarogenesis Assay on Matrigel^®^

MMECs were seeded (4 × 10^4^) on 48-well plates coated with Matrigel^®^ (BD Biosciences) in serum-free media (SFM) (negative control), in SFM supplemented with VEGF and FGF2 (both 10 ng/mL; Miltenyi Biotech, Cologne, Germany) (positive control) or with FBEVs. Skeletonization of the mesh was detected using a brightfield microscope coupled with a high-resolution camera (EVOS, Thermo Fisher Scientific) at 4× and 10× magnification after 3, 6, 12, and 24 h. Measurement of vessel length and branching points was performed using EVOS software (v1.5.1355.293) in three randomly chosen fields.

### 2.8. Supernatant Preparation

Cell supernatants from MMECs (3 × 10^5^) co-cultured for 24 h with/without FBEVs were centrifuged at 450 g at 4 °C for 5 min to eliminate cell debris and concentrated at 1 × 10^6^ cells/mL by centrifugation for 60 min at 4000× g at 4 °C. Concentrated cell supernatants were used for angiogenesis array or zymography assay.

### 2.9. Protein Extraction

The MMECs and FBEVs were resuspended in RIPA buffer (Thermo Fisher Scientific) containing a cocktail of protease and phosphatase inhibitors (Merck, Rove, NJ, USA) at a final concentration of 1%. Cell lysis was performed on ice in agitation for 40 min. Lysates were then centrifuged at 15,000× *g* for 15 min at 4 °C. Protein concentration was assessed by using Bradford assay (Biorad).

### 2.10. Angiogenesis Array

Protein lysates (600 μg) from FBEVs or cell supernatant from MMECs (3 × 10^5^) were analyzed using the Human Angiogenesis Array (Biotechne). The assay was performed according to the manufacturer’s instructions. Densitometric quantification of the resulting membranes was performed using Kodak Molecular Imaging Software 5.0 (Eastman Kodak Co., Rochester, NY, USA). Each protein’s average pixel density was normalized to reference spots.

### 2.11. Phospho-Kinase Array

Cell lysates (200 μg) from MMECs (5 × 10^5^) co-cultured with/without FBEVs for 6 or 24 h were analyzed using a Human Phospho–Kinase Array (Biotechne) according to the manufacturer’s instructions. Densitometric quantification of the resulting membranes was performed using Kodak Molecular Imaging Software (Eastman Kodak Co) and the average pixel density of each protein was normalized to reference spots.

### 2.12. Phospho-Receptors Analysis

The MMECs were co-cultured with/without FBEVs for 1 h, fixed and permeabilized using Cytofix/Cytoperm™ (BD Biosciences) and stained with rabbit anti-human monoclonal antibodies against phospho-VEGFR2, phospho-HGFR and phospho-Tie2 (Biotechne) followed by staining with PE-labeled anti-rabbit IgG antibody (BD Biosciences). Samples were acquired using FACScanto II (BD Biosciences, Franklin Lakes, NJ, USA). Data analysis was performed with FlowJo v.10.

### 2.13. Scratch Assay

The MMECs (4 × 10^4^) were seeded in 24-well plates to form a cell monolayer. The day after, a scratch was made by scraping the cell monolayer with a sterile pipette tip, and cells were cultured in SFM (negative control) to minimize the contribution of cell proliferation, DMEM supplemented with 1.5% exosome-depleted FBS and VEGF/FGF2 10 ng/mL (Miltenyi Biotech) (positive control) or DMEM supplemented with FBEVs. Twenty-four hours later, MMECs were fixed and stained with Crystal violet. The number of migrated cells was determined by counting the MMECs into the “scratch” (migrated cells/field).

### 2.14. Chemotaxis Assay

The MMECs (3.5 × 10^4^) co-cultured with/without FBEVs for 6 or 24 h were tested in a Boyden chamber on a polycarbonate membrane (Neuro Probe, Inc. Gaithersburg, MD, USA) pre-coated with 10 µg/mL fibronectin (Merck) using SFM as negative control and DMEM supplemented with 1.5% exosome-depleted FBS and VEGF and FGF2 10 ng/mL (Miltenyi Biotech) as chemoattractant factors. After 24 h at 37 °C, the migrated cells were fixed, stained, and counted in at least three randomly chosen fields by the EVOS inverted microscope (Thermo Fisher Scientific) at 40×.

### 2.15. Zymography

Concentrated cell supernatants were mixed with sodium dodecyl sulphate buffer under non-reducing conditions and run on Novex^®^ Zymogram gelatin gel (Thermo Fisher Scientific) at 125 V for 90 min. After electrophoresis, the enzyme was renatured by incubating the gel in Zymogram renaturing buffer containing a non-ionic detergent. The gel was equilibrated in Zymogram developing buffer and then stained and de-stained according to the manufacturer.

### 2.16. Real-Time Quantitative RT-PCR (RT-qPCR)

Total mRNA was isolated from MMECs using the RNeasy Micro kit (Qiagen, Milan, Italy) and reverse-transcribed with the iScript cDNA Synthesis Kit (BioRad, Bio-rad, Hercules, CA, USA) according to the manufacturer’s instructions. Gene expression was analyzed using SsoAdvanced Universal Probes Supermix (Bio-Rad Laboratories, Bio-rad, Hercules, CA, USA) and specific TaqMan assays (Gapdh: Hs02758991_g1, VegfA: Hs00900055_m1, MMP2: Hs01548727_m1, IL6: Hs00985639_m1, IL8: Hs00174103_m1). Relative gene expression was normalized to *GAPDH* as endogenous control, and the relative fold-change was measured using the 2^−ΔΔCt^ formula.

### 2.17. Statistics

This was performed using GraphPad Prism5 software (GraphPad Software, LLC.). Data are presented as mean±standard deviation (SD). An unpaired Mann–Whitney U-test was performed when two datasets were compared. One-way ANOVA with Tukey’s multiple comparisons was used to compare three or more datasets. *P* < 0.05 was considered statistically significant.

## 3. Results

### 3.1. FBEVs Promote In Vitro Angiogenesis

FBEVs were purified from cultured FBs from MM patients and characterized as previously described [15]. In particular, FBEVs were 160 nm in size, as shown in Appendix A.

To determine the effect of FBEVs on MMECs, we first performed a time-dependent analysis of EVs uptake by MMECs using flow cytometry.

As shown in Figure 1A,B, MMECs were virtually negative for BODIPY TR™ ceramide-labelled Evs at 3 and 6 h of co-culture, while they became positive at 12 and 24 h, indicating FBEVs internalization. Accordingly, dual confocal immunofluorescence analysis (Figure 1C) proved the uptake of labelled FBEVs in MMECs only at 12 and 24 h of coculture. Surprisingly, a time-dependent analysis of the capillarogenesis assay showed that FBEVs exposure triggered in vitro angiogenesis already at 3 and 6 h of co-culture (Figure 1D). In detail, after 3 h FBEVs induced the formation of cell protrusions and early vessels that culminate at 6 h when FBEVs-treated MMECs showed increased capillary-like structures with significantly enhanced relative vessel length and branching points compared to untreated (negative control) and VEGF/FGF2-treated (positive control) cultures (Figure 1D,E).

Overall, these results suggest that FBEVs exert an early pro-angiogenic effect without a detectable uptake.

### 3.2. FBEVs Contain Angiogenic Cytokines

As recent literature data have shown that EVs can exert their activity in the extracellular space [17,18], we analyzed FBEVs content using an angiogenesis array to investigate the mechanisms involved in their early pro-angiogenic effect.

Our analysis demonstrated that FBEVs were enriched in the canonical pro-angiogenic factors VEGF, HGF, and ANG-1. They also contained low levels of anti-angiogenic cytokines (i.e., ANG2, Serpin B5, TIMP4, and TPS-2) (Figure 2A and Appendix A). To test whether the identified cytokines were responsible for the early FBEVs angiogenic effect, we evaluated the effect of FBEVs on the phosphorylation status of the VEGF, HGF, and ANG-1 receptors (VEGFR2, HGFR, and Tie2, respectively), which represents an early event of intracellular signaling activation. Flow cytometric analysis of MMECs treated with FBEVs for 1 h showed an increase in VEGFR2, HGFR, and Tie2 phosphorylation (Figure 2B), suggesting a cytokine-mediated activation. In line with this evidence, MMECs co-cultured with FBEVs for 1 h acquired an activated phenotype (star-like shape on Matrigel^®^-based 3D culture) and increased spreading (Figure 2C). In addition, pre-treatment of MMECs with blocking antibodies against VEGFR2, HGFR, and Tie2 receptors dampened FBEVs induced activation (Figure 2D). Interestingly, while the blockade of VEGFR2 was sufficient to significantly reduce FBEVs-induced MMECs spreading, blocking HGFR and Tie2 only showed partial attenuation (Figure 2D,E).

These results suggest that FBEVs foster an early uptake-independent angiogenic effect in a cytokine-mediated fashion.

### 3.3. FBEVs Induce a Late Angiogenic Response

We then investigated the pro-angiogenic effect of FBEVs on MMECs after their uptake. To this purpose, MMECs were co-cultured with FBEVs for 24 h and analyzed for several angiogenic functions: migration, chemotaxis, invasion, and capillarogenesis.

As shown in Figure 3A, MMECs co-cultured with FBEVs acquired increased migratory capabilities compared to untreated MMECs (negative control) and to MMECs cultured with VEGF/FGF2 (positive control) (Figure 3A). This finding was also confirmed by chemotaxis assays; indeed, MMECs exposed to FBEVs showed enhanced migration toward VEGF/FGF2 as well as toward SFM (Figure 3B).

Next, we evaluated the involvement of FBEVs in MMECs’ invasion ability by analyzing the production and the release of matrix metalloproteinases (MMPs). RT-qPCR analysis showed that MMECs co-cultured with FBEVs increased MMP-2 expression (Figure 3C). Moreover, FBEVs stimulated the secretion of both MMP-2 and MMP-9 by MMECs because of the cleavage of their inactive forms, pro-MMP2 and pro-MMP9, respectively (Figure 3D). Finally, pre-treatment for 24 h with FBEVs enhanced the ability of MMECs to form tubes with multicentric junctions and to originate a meshwork of capillary-like structures on Matrigel^®^ (Figure 3E).

These results demonstrate that FBEVs induce an over-angiogenic activity of MMECs also after their uptake.

### 3.4. FBEVs Modulate Intracellular Pathways and Induce the Release of Pro-Angiogenic Cytokines

To identify the downstream pathways involved in the uptake-independent/-dependent pro-angiogenic effects of FBEVs, we explored the activation status of MMECs signaling pathways after co-culture with FBEVs for 6 and 24 h using a phosphor–kinase array.

As shown in Figure 4A,B, phosphoproteomic analysis highlighted a different pattern of signaling activation. Activation of the PI3K/AKT/mTOR pathway was observed both at 6 and 24 h. However, co-culture of MMECs with FBEVs for 6 h significantly increased the levels of phospho-AKT (T308 and S473) but not of phospho-p70S6K, implying the activation of mTORC2 pathway, involved in MMECs migration and actin reorganization [19]. In contrast, FBEVs internalization activated mTORC1 pathway in MMECs, as shown by the phosphorylation status of p70S6K (T389) and AKT (T308). Furthermore, AKT activation promoted the phosphorylation of other downstream targets, including GSK-3, CREB, eNOS, and p53. Comparison of Figure 4A with Figure 4B highlighted an increased activation of MAPK, SRC, and STAT pathways in MMECs co-cultured with FBEVs for 24 h. Activation of Wnt/β-catenin was instead observed only at 6 h of co-culture. Accordingly, RT-qPCR analysis of β-catenin target genes [20,21,22] showed a significant increase in VEGF, IL-6, and IL-8 mRNA levels in MMECs co-cultured with FBEVs for 6 h (Figure 4C,D).

Finally, as the signaling pathways modulated by FBEVs also regulate cytokine production [23,24], we investigated whether they could affect the release of angiogenic cytokines by MMECs. Remarkably, MMECs co-cultured with FBEVs for 24 h modified their secretory profile by inducing the secretion of pro-angiogenic factors (VEGF, ANG-1, HGF, and IGFBP-1) and by decreasing the release of anti-angiogenic ones (PXT-3, TIMP-1, and THBS-1) (Figure 4E). Overall, our results demonstrate that FBEVs modulate the release of specific cytokines that additionally support angiogenesis in the BM microenvironment of MM patients.

## 4. Discussion

In the BM microenvironment, MM cells and BMSCs establish a mutual crosstalk that sustains the formation of a pro-survival niche [2,25]. We have previously demonstrated that MM cells activate MM FBs via TGF-β and EVs release and reprogram their miRNA expression profile [12,26]. On the other hand, MM FBs sustain MM cell proliferation, survival, and drug resistance through cytokines and EVs [12,15,27]. In this study, we investigated the ability of MM FBs to shape the BM pro-angiogenic niche.

Tumor angiogenesis is a tightly regulated process triggered by several mechanisms, including an unbalanced equilibrium between pro- and anti-angiogenic factors [26]. Recently, EVs have emerged as key mechanism able to promote BM angiogenesis [11].

Despite the growing literature data on EVs, most studies have investigated their ability to affect recipient cell behavior exclusively after their uptake and cargo transfer. Wang et al. have demonstrated that mice BMSCs enhance BM angiogenesis via EVs release by delivering several angiogenesis-related proteins and activating intracellular STAT3, c-Jun N-terminal kinase, and p53 pathways in recipient cells [26]. Further studies indicated that MM-EVs contain CD147 [28], piRNA-823 [29], miR-340, and miR-135b [30,31], which may support their pro-angiogenic effect.

In this study, we analyzed the involvement of FBEVs in BM angiogenesis and we demonstrated that FBEVs exert a pro-angiogenic effect on MMECs in a two-steps process: an early uptake-independent activation phase occurring shortly after EVs exposure (cell spreading at 1–3 h), and a late uptake-dependent effector phase after longer EVs exposure (tube formation, peak at 24 h). Specifically, co-culture of MMECs with FBEVs for 1 h triggered MMECs spreading that leads to increased protrusions and cell junctions at 3 h and culminates in the formation of an intricate network at 6 h.

The analysis of FBEVs’ cargo revealed an enrichment in angiogenic cytokines (i.e., ANG-1, VEGF, and HGF), and the use of blocking antibodies against their cognate receptors dampened the pro-angiogenic effect of FBEVs. Although the inhibition of these receptors was sufficient to attenuate FBEVs-induced angiogenesis, it did not entirely abrogate MMECs activation, suggesting that other factors are involved to a smaller extent in the pro-angiogenic effect. For instance, FBEVs contain many other cytokines (i.e., angiogenin, FGF-7, IL-8, MCP-1, MMPs, uPA, and PIGF), ncRNAs, and proteins that all collectively contribute to foster angiogenesis, leading to resistance and low response to angiogenic inhibitor [28,29,30,31,32,33,34,35].

Cytokines can be presented in a membrane-bound form on EVs surface or encapsulated into vesicles as cargo [36]. For example, TNFα exists in a membrane-bound form on the EVs released by synovial fibroblasts in rheumatoid arthritis patients [37], exerting its biologically active function. Other cytokines have been reported to exist as biologically active in both soluble and membrane-bound forms. Interestingly, several isoforms of VEGF have been described to have multiple localization, including the association with the plasma membrane [38]. Recently, Toth et al. demonstrated the presence of proteins corona on EVs surface that equips EVs with an additional internalization-independent function [39]. In line with this hypothesis, several studies have shown that the initial interaction between EVs and their target cell involves physical contact through different surface molecules that activate intracellular signaling pathways [40]. This event, preceding EVs internalization, has already been described for antigen presentation and anchorage-independent tumor growth [41,42]. Additional studies have also shown that in the peripheral blood, EVs establish dynamic interactions with ECs. Indeed, after rolling, they arrest on ECs membrane where they accumulate for an average of 3 min, suggesting the existence of a potential interaction between EVs and specific receptors on ECs [43].

The cytokine cargo of FBEVs could also be released in the extracellular space. A similar process has been described for tumor-derived EVs that become leaky after perforin secretion by cytotoxic T cells [44] and for eosinophil-derived granules that release cytokines [45]. Further studies are needed to better identify the short-term mechanisms through which FBEVs cytokines activate MMECs.

The dual pro-angiogenic effect of FBEVs differentially affects intracellular signaling: with a short time exposure, we observed the activation of mTORC2 pathway, while with a longer time exposure (maximal EVs internalization), we observed the induced mTORC1 activation together with other downstream proteins such as GSK-3, CREB, eNOS, and p53. The different modulation of signaling cascades at early/late time points is associated with different transcriptional programs (early: β-catenin, late: STATs) that, in turn, prompt a late release of pro-angiogenic cytokines, including VEGF, HGF, and ANG-1, implying the creation of an autocrine pro-angiogenic loop.

Hence, our data suggest that FBEVs are an additional source of pro-angiogenic factors (i.e., VEGF, FGF2, and ANG-1) with implications for patients’ outcome and response to therapy. The analysis of circulating angiogenic cytokines in MM patients enrolled in the GIMEMA MM0305 trial has shown that higher levels of VEGF and FGF2 are associated with lower overall survival and progression-free survival [46]. Increased amounts of ANG-2, FGF2, HGF, IL-8, PDGF-BB, TNF-α, and VEGF are signs of poor response [46].

Response to therapy represents an important and still unsolved problem for anti MM drugs, including the anti-angiogenic ones. Indeed, despite the promising preclinical studies, anti-VEGF therapies (i.e., anti-VEGF monoclonal antibodies or dual inhibition of VEGF/cMET) have shown little efficacy in vivo without significantly improving MM patients’ outcome [47]. The EVs’ pro-angiogenic cargo and the EVs-mediated feedback loops of cytokines within the BM milieu could be an additional mechanism of the anti-angiogenic therapy escape in MM patients.

Overall, our results demonstrate that EVs create a pro-angiogenic milieu through a dual time-related mechanism: an early uptake-independent mechanism and a late uptake-dependent mechanism that induce the activation of different intracellular pathways and transcriptional programs which have implications for the design of novel anti-angiogenic strategies in MM.

## Figures and Tables

**Figure 1 biomedicines-11-01400-f001:**
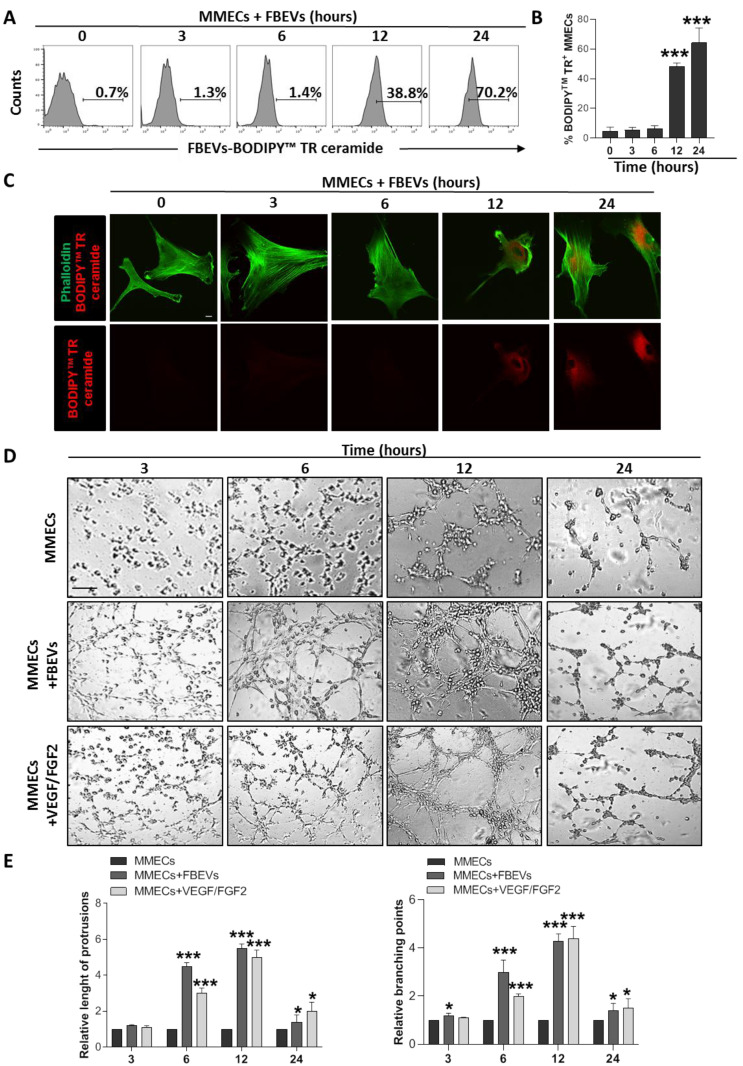
FBEVs induce an early uptake-independent over angiogenic effect in MMECs. (**A**) MMECs (n = 5) were co-cultured with BODIPY TR™ ceramide labeled FBEVs and analyzed by flow cytometry at different time points (0, 3, 6, 12, and 24 h). Representative analysis of time-dependent uptake of Evs is shown. (**B**) Bar graphs represent the percentage values of BODIPY TR™ positive MMECs. Data are expressed as mean ± S.D. (**C**) MMECs were co-cultured for 0, 3, 6, 12, and 24 h with FBEVs labelled with BODIPY TR ceramide (red), specific for cell membranes, and stained for F-actin fibers (green). Representative images of the confocal dual immunofluorescence images are shown. Scale bar = 10 μm. (**D**) MMECs were cultured alone, with VEGF/FGF2 (as positive control) or co-cultured with FBEVs on Matrigel^®^-coated 48-well plates and tested for in vitro angiogenesis at 3, 6, 12 and 24 h. Representative images of five independent in vitro angiogenesis assays are shown. Scale bars = 50 μm. Original magnification: ×200. (**E**) Bar graphs represent relative length of protrusions and branching points in MMECs co-cultured with serum free media (SFM), FBEVs or VEGF/FGF2. Data are expressed as mean ± S.D. * *p* < 0.05, *** *p*< 0.001 vs. MMECs.

**Figure 2 biomedicines-11-01400-f002:**
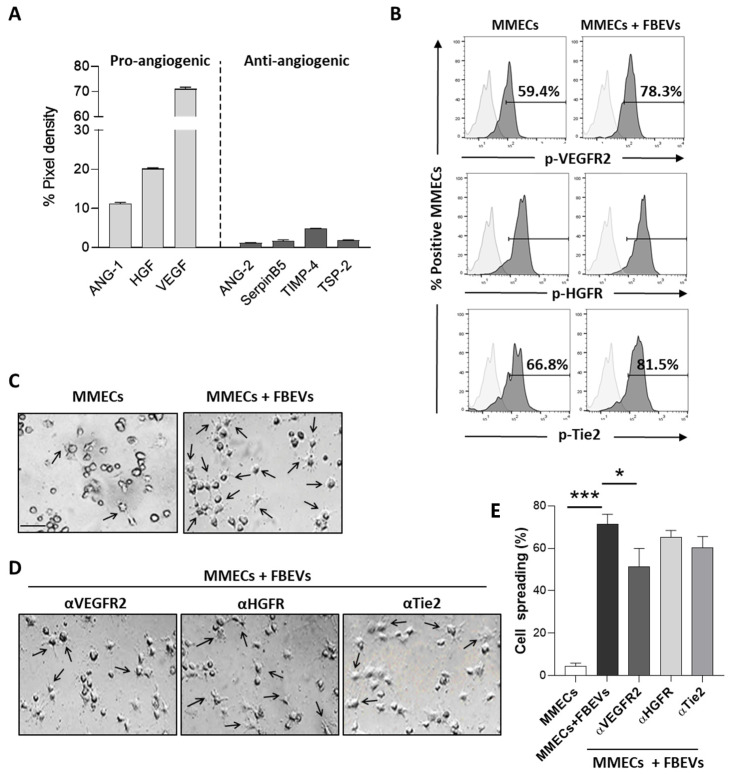
FBEVs contain pro-angiogenic cytokines and activate MMECs. (**A**) FBEVs were lysed and analyzed for cytokine cargo using an angiogenic array. Data are expressed as pixel density, normalized to reference spots, and expressed as mean ± S.D. (**B**) Representative flow cytometry analysis of MMECs co-cultured with FBEVs for 1 h and analyzed for phospho-VEGFR2, phospho-HGFR, and phospho-Tie2 expression by flow cytometry. (**C**) Analysis of cell sprouting of MMECs cultured with/without FBEVs and/or (**D**) pre-treated with blocking antibodies against VEGFR2, HGFR, and Tie2. Representative images of three independent experiments of Matrigel^®^-based 3D culture are shown. Scale bar = 25 μm. (**E**) Bar graphs indicate the percentage of cell spreading/field of MMECs cultured with/without FBEVs and/or pre-treated with blocking antibodies against VEGFR2, HGFR, and Tie2. Data are expressed as mean ± S.D. * *p* < 0.05, *** *p* < 0.001.

**Figure 3 biomedicines-11-01400-f003:**
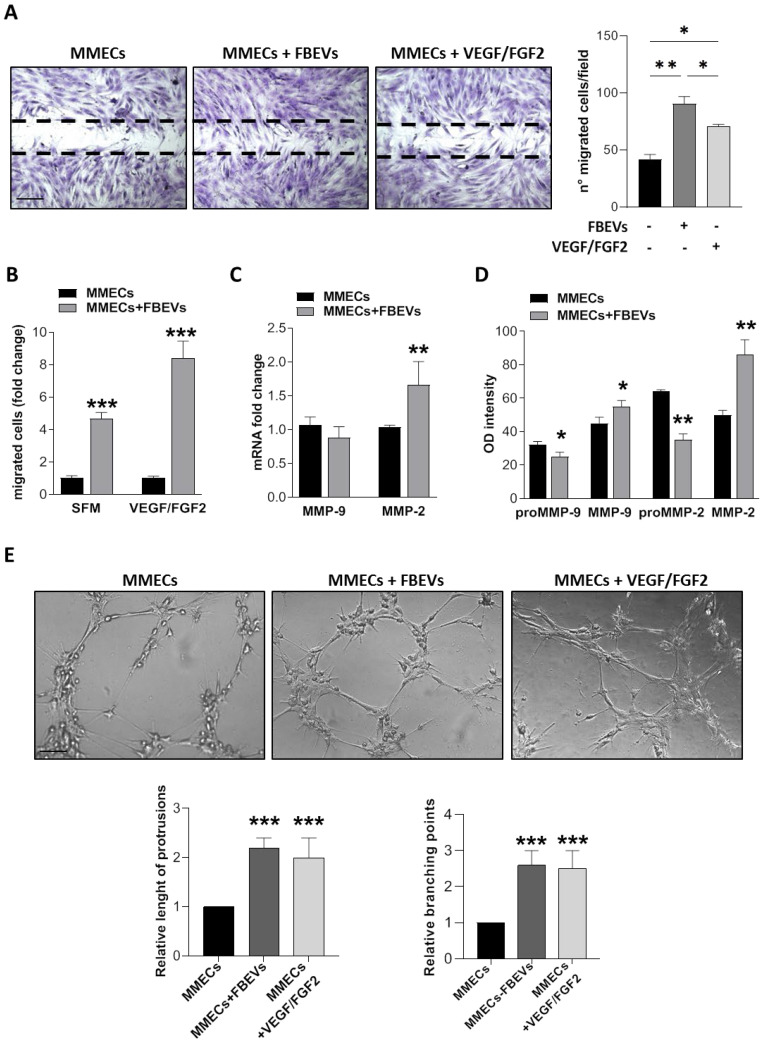
FBEVs induce a late over-angiogenic response in MMECs. MMECs were co-cultured with FBEVs for 24 h and tested for different angiogenic assays (n = 5 different MM patients). (**A**) Spontaneous migration using the scratch assay. Representative images of scratch closure 16 h after the scratch. Scale bar = 50 μm. Original magnification: ×200. Bar graphs represent the number of migrated cells/field expressed as mean ± S.D. (**B**) Chemotaxis toward serum free medium (SFM) or chemoattractive medium (VEGF/FGF2). Data are expressed as a mean of relative migrated cells/field ± S.D. (**C**) MMP-2 and MMP-9 mRNA expression was analyzed by RT-qPCR. mRNA fold-expression was normalized to endogenous GAPDH and expressed as mean ± S.D vs. MMECs. (**D**) Conditioned media of MMECs alone or co-cultured with FBEVs were tested for zymography to determine the amount of active MMP-2 and MMP-9. Bar graphs represent OD intensity of the protease activity regions. (**E**) Representative images of in vitro angiogenesis assay of MMECs seeded on Matrigel^®^-coated 48-well plates after 16 h (n = 3 independent experiments). Scale bar = 25 μm. Bar graphs represent the relative length of protrusions and branching points in MMECs co-cultured with FBEVs vs. MMECs alone. Data are expressed as mean ± S.D. * *p* < 0.05, ** *p* < 0.005, *** *p* < 0.001.

**Figure 4 biomedicines-11-01400-f004:**
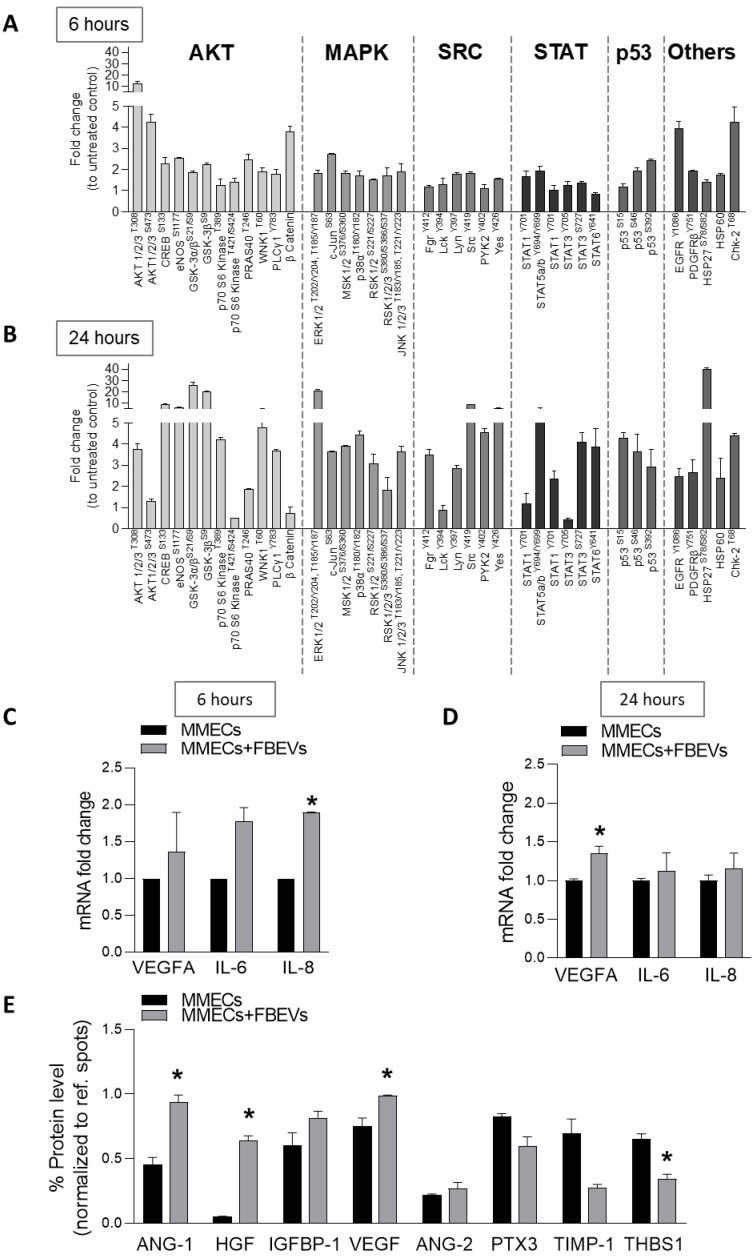
FBEVs modulates intracellular pathways in MMECs. MMECs were cultured alone or co-cultured with FBEVs (n = 3 patients) for (**A**,**C**) 6 h or (**B**–**E**) 24 h. (**A**,**B**) Cell lysates were analyzed using a phospho-kinase array. (**C**,**D**) VEGFA, IL-6 and IL-8 mRNA levels were analyzed byRT-qPCR. mRNA fold-expression was normalized to endogenous GAPDH and expressed as mean ± S.D. vs. MMECs. (**E**) Conditioned media of MMECs cultured alone or co-cultured with FBEVs for 24 h was tested for angiogenic cytokines using an angiogenesis array. Data are normalized to reference spots and expressed as mean protein levels ± S.D. * *p* < 0.05.

## Data Availability

Not applicable.

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
