# Peer review of "Uptake-Dependent and -Independent Effects of Fibroblasts-Derived Extracellular Vesicles on Bone Marrow Endothelial Cells from Patients with Multiple Myeloma: Therapeutic and Clinical Implications"

_biomedicines, 2023, doi:10.3390/biomedicines11051400_

Round 1
Reviewer 1 Report
The authors studied the influence of EVs in marrow environment of myeloma patients and showed potential angiogenesis pathway of myeloma. It is an interesting issue with well prepared text.
Author Response
Author's Reply to the Reviewer 1 Report
Reviewer 1
The authors studied the influence of EVs in marrow environment of myeloma patients and showed potential angiogenesis pathway of myeloma. It is an interesting issue with well prepared text.
The authors thank the Reviewer 1 for positive comments.

Reviewer 2 Report
The authors demonstrated that fibroblast-derived exosomes from myeloma patients promote angiogenesis by cytokines. This article is poor in novelty because numerous articles which described that the exosome from myeloma cells and their microenvironments affects angiogenesis. Moreover, the following issues are necessary to be described. Therefore, this article cannot be published in our journal.
1. The authors should demonstrate the comparison of the effects between fibroblast-derives exosomes from myeloma patients and healthy persons.
2. The authors should uncover the education of fibroblasts from myeloma cells or other cells in the myeloma microenvironment.
3. The definition of vessels is necessary for the experiments of capillarogenesis.
Author Response
Author's Reply to the Reviewer 2 Report
Reviewer 2
The authors demonstrated that fibroblast-derived exosomes from myeloma patients promote angiogenesis by cytokines. This article is poor in novelty because numerous articles which described that the exosome from myeloma cells and their microenvironments affects angiogenesis. Moreover, the following issues are necessary to be described. Therefore, this article cannot be published in our journal.
Reply. We agree with the Reviewer’s comment that several studies have demonstrated the effect of exosomes from myeloma cells and bone marrow stromal cells on angiogenesis. Accordingly, in the discussion section we have quoted these studies (references 25-31). Nevertheless, no study has analysed the pro-angiogenic effect of exosomes from bone marrow fibroblasts. In addition, all these studies demonstrated that the pro-angiogenic effect of microvesicles was related to their uptake from recipient cells.
The novelty of our study is the demonstration that FBEVs play an early proangiogenic effect in the absence of their uptake: “FBEVs foster MM angiogenesis through a dual time-related uptake-independent and uptake-dependent mechanisms that activate different intracellular pathways and transcriptional programs” (Abstract, lines 34-36). In particular, “we demonstrate that FBEVs cargo contains several angiogenic cytokines (i.e., VEGF, HGF, and ANG-1) that promote an early over-angiogenic effect independent from EVs uptake”. They activate “the VEGF/VEGFR2, HGF/HGFR, and ANG-1/Tie2 axis, as well as mTORC2 and Wnt/β-catenin pathways, suggesting that the early over-angiogenic effect is a cytokine-mediated process”. Based on these results, we suppose that “FBEVs are an additional source of pro-angiogenic factors (i.e., VEGF, FGF2, ANG-1) with implications for patients’ outcome and response to therapy and provide the rationale for designing novel anti-angiogenic strategies”.
Reviewer’s comment 1: The authors should demonstrate the comparison of the effects between fibroblast-derives exosomes from myeloma patients and healthy persons.
Reply. The reviewer’s comment is intriguing but it is hard to realize for several reasons and in primis for the difficulty to obtain a significant number of bone marrow biopsies from healthy people. In addition, normal fibroblasts are detected at low percentage values in the bone marrow and are characterized by a low proliferative index and a high susceptibility to apoptosis [Frassanito MA et al., Leukemia 2014; Kalluri R and Zeisberg M, Nature 2006]. For these reasons, despite the intriguing question we are not able to perform this issue.
Reviewer’s comment 2: The authors should uncover the education of fibroblasts from myeloma cells or other cells in the myeloma microenvironment.
Reply. In the BM microenvironment MM cells and BMSCs co-evolve during MM progression by establishing a mutual cross-talk that “educate” the BM to behave as a pro-survival niche. In this context, MM cells shape BMSCs via cytokines, miRNA, aberrant pathways activation and, more recently, via extracellular vesicles release. In turn, the BMSCs ensure MM cells survival and proliferation and contribute to transform neighbouring BM cells. We have previously demonstrated that MM cells “educate” FBs in different ways. Indeed, MM cells actively secrete TGF-β that induces FB activation and promotes their proliferation and recruitment via Endothelial-to-Mesenchymal transition [Frassanito MA et al., Leukemia 2014]. Furthermore, EVs derived from MM cells (MM-EVs) increase the expression of fibroblast activation protein (FAP) and alpha-smooth muscle actin (αSMA) suggesting their ability to activate FBs and promote their transformation into cancer associated fibroblasts [Frassanito MA et al., J Pathol, 2016]. MM-EVs are also able to reprogram miRNA expression profile of recipient FBs by inducing a de novo miRNA synthesis though the activation of Hippo pathway [Frassanito MA et al., J Pathol, 2016].
According to reviewer’s suggestion, we have now summarized the complex mechanism of fibroblast education in the BM niche by adding a sentence in the Discussion section: “In the BM microenvironment MM cells and BMSCs establish a mutual crosstalk that sustains the formation of a pro-survival niche [2,25]. We have previously demonstrated that MM cells activate MM FBs via TGF-β and EVs release and reprogram their miRNA expression profile [12,26]. On the other hand, MM FBs sustain MM cell proliferation, survival, and drug resistance through cytokines and EVs [12,15,27]. In this study, we have investigated the ability of MM FBs to shape the BM pro-angiogenic niche. Tumor angiogenesis is a tightly regulated process triggered by several mechanisms, including an unbalanced equilibrium between pro- and anti-angiogenic factors [26]. Recently, EVs have emerged as key mechanism able to promote BM angiogenesis [11]. Despite the growing literature data on EVs, most studies have investigated their ability to affect recipient cell behavior exclusively after their uptake and cargo transfer.” (page 13 lines 347-357)
Reviewer’s comment 3: The definition of vessels is necessary for the experiments of capillarogenesis.
Reply: In the capillarogenesis assay, ECs seeded onto a basement membrane-like surface (e.g., Matrigel®) form a network of capillary-like structures, which recapitulate angiogenesis. Indeed, ECs spread throughout the matrigel surface and aligned to form branching, anastomosing and thick tubes with multicentric junctions [Vacca A et al., Blood 2003]. The term ‘length of vessels’ means the length of the protrusion between two branching point. However, as this term can be misleading, we changed it with “length of protrusions”.

Reviewer 3 Report
Congratulations for your aticle. It is interesting about therapeutic and clinical implications.
I recommended the minor changes as are:
1- In the EVs purification and co-culture section, you should be explain briefly the reference [15].
2. Figures have poor resolution, you should improve.
Author Response
Author's Reply to the Reviewer 3 Report
Reviewer 3
Congratulations for your aticle. It is interesting about therapeutic and clinical implications.I recommended the minor changes as are:
1- In the EVs purification and co-culture section, you should be explain briefly the reference [15].
- Figures have poor resolution, you should improve.
The authors thank the Reviewer 3 for helpful criticism and are glad for positive comments.
Reviewer’s comment 1. In the EVs purification and co-culture section, you should be explain briefly the reference [15].
Reply. According to reviewer’s suggestion, in the EVs purification and co-culture section we added new sentences that explain the method as follow: “The MM FBEVs were purified from MM FBs cultured for 72 hours in DMEM medium (Sigma-Aldrich) supplemented with 10% of exosome-depleted FBS (Euroclone) as described [1]. Briefly, cell culture media were sequentially centrifuged at increasing speeds to remove dead cells and debris, filtered with 0.22-μm pore filters (Merck Millipore, Darmstadt, Germany) to remove small debris and larger vesicles, and concentrated with a protein concentrator (Ultra-15 MWCO 10 kDa; Merck Millipore). EVs were precipitated using ExoQuick-TC exosome precipitation solution (SBI System Biosciences, Palo Alto, CA, USA) according to the manufacturer’s instructions. Purified FBEVs were characterized by flow cytometry using the Megamix-Plus SSC kit (BioCytex, Marseille, France) according to the manufacturer’s instructions. The protein concentration of each EVs pellet was quantified using the Bradford assay (Bio-Rad Laboratories, Hercules, CA, USA). Co-culture experiments were performed by adding 400 µg of FBEVs to 4x104 MMECs.” (page 3 lines 105-116).
Reviewer’s comment 2. Figures have poor resolution, you should improve.
Reply. We are very sorry for the poor resolution of the Figures. We separately uploaded the high-resolution figures as TIFF files but we are not able to paste the high resolution figures in the final merged version of our revised manuscript.

Reviewer 4 Report
The present study assess the angiogenic effects of myeloma FB-EV in bone marrow myeloma endothelial cells. The manuscript is very well written, structured and accomplishes its main goal of investigating the mechanisms that govern FB-EV mediated angiogenesis. However, this reviewer thinks that with few amendments, this manuscript would gain robustness and impact:
11) Title: In its current form, the title does not reflect the study fully and is not specific enough. “Early and late effects” is very vague for the level of detail that the authors reach in this study, another better term, already used in the text, would be: “Uptake-independent and uptake-dependent”. Also, rather than “in patients” it should be “from patients”, otherwise the title implies studies in humans, when the study is fully carried in vitro.
22) The uptake results are critical for the rest of the paper. However, authors only use flow cytometry to assess this uptake. The manuscript should include a Nano-Tracking Analysis and or SEM/TEM, in order to understand the characteristics of the vesicles. Also live-imaging of this uptake, which would add credibility to flow cytometry results. The fast increase between 6 and 12 hours could be due to diffusion of the dye between lipidic molecules, and other studies have shown a much faster uptake of EV than 12 hours. Therefore, imagining analysis is required.
33) The wound healing / scratch assay does not include an agent to prevent proliferation, so therefore the results extrapolated cannot be only be explained through migration but also proliferation. This should be discussed. Alternatively, the effect on proliferation could be assessed with a simple cell-based assay.
44) Discussion, page 11, line 335: Authors acknowledge than the inhibitors were not sufficient to prevent angiogenesis in the present results. Could authors include references of other studies where this phenomenon is observed? This would further support their theory of alternative mechanisms to FB-EV angiogenesis.
Minor comments:
55) Figures have very poor resolution and are difficult to read, please amend.
66) Wound healing assay should be renamed to “scratch assay”
77) Please include scale bars in all images from microscopy
88) Figure 2. The Last bar graph should have a letter to indicate its description in the legend. In this form is confusing.
9) 3.1. Results. Could authors reference another study(s) that report these exhaustion of this cell type after 24 hours? Seems a short time point to this reviewer
Round 2
Reviewer 2 Report
The reviewer has read the revised version very carefully. The authors have responded to the comments by the reviewers adequately. Therefore, the reviewer has no additional revision. This revised version in the present form is acceptable to our journal.
Reviewer 4 Report
Authors have addressed all the points raised by this reviewer.